

# Recent improvements of Long-Path DOAS measurements: impact on accuracy and stability of short-term and automated long-term observations

Jan-Marcus Nasse[1], Philipp G. Eger[1, 2], Denis Pöhler[1], Stefan Schmitt[1], Udo Frieß[1], and Ulrich Platt[1]

[1]Institute of Environmental Physics, University of Heidelberg, Im Neuenheimer Feld 229, D-69120 Heidelberg, Germany
[2]now at: Max Planck Institute for Chemistry, Hahn-Meitner-Weg 1, D-55128 Mainz, Germany

*Correspondence to:* J.-M. Nasse (jan.nasse@iup.uni-heidelberg.de)

**Abstract.** Over the last decades, Differential Optical Absorption Spectroscopy (DOAS) has been used as a common technique to simultaneously measure abundances of a variety of atmospheric trace gases. Exploiting the unique differential absorption cross section of trace gas molecules, mixing ratios can be derived by measuring the optical density along a defined light path and by applying the Beer-Lambert law. Active long-path (LP-DOAS) instruments can detect trace gases along a light path of

a few hundred metres up to $20\,km$ with sensitivities for mixing ratios down to ppbv and pptv levels, depending on the trace gas species. To achieve high measurement accuracy and low detection limits, it is crucial to reduce instrumental artefacts that lead to systematic structures in the residual spectra of the analysis. Spectral residual structures can be introduced by most components of a LP-DOAS measurement system, namely by the light source, in the transmission of the measurement signal between the system components or at the level of spectrometer and detector. This article focuses on recent improvements by the

first application of a new type of light source and consequent changes to the optical setup to improve measurement accuracy.

Most state-of-the-art LP-DOAS instruments are based on fibre optics and use xenon arc lamps or light emitting diodes (LEDs) as light sources. Here we present the application of a Laser Driven Light Source (LDLS), which significantly improves the measurement quality compared to conventional light sources. In addition the lifetime of LDLS is about an order of magnitude higher than of typical Xe-arc lamps. The small and very stable plasma discharge spot of the LDLS allows the application

of a modified fibre configuration. This enables a better light coupling with higher light throughput, higher transmission homogeneity, and a better suppression of light from disturbing wavelength regions. Furthermore, the mode mixing properties of the optical fibre are enhanced by an improved mechanical treatment. The combined effects lead to spectral residual structures in the range of $5 - 10 \cdot 10^{-5}$ RMS (in units of optical density). This represents a reduction of detection limits of typical trace gas species by a factor of 3-4 compared to previous setups. High temporal stability and reduced operational complexity of this new

setup allow the operation of low-maintenance automated LP-DOAS systems as demonstrated here by more than two years of continuous observations in Antarctica.



# 1 Introduction

Active long-path differential optical absorption spectroscopy (LP-DOAS) is a well-established remote sensing technique based on the DOAS principle introduced by Perner et al. (1976); Platt and Perner (1980, 1983). It can attain detection limits in the order of ppbv to pptv (nanomole per mole to picomole per mole) for absorbers in the ultraviolet to near-infrared spectral

range (270-800 nm). Detectable species include $NO_2$, $O_3$, $SO_2$, ClO, OClO, BrO, IO, OBrO, OIO, $I_2$, OIO, formaldehyde, glyoxal, and the oxygen dimer $O_4$. LP-DOAS setups have been used in various applications such as studying urban pollution (Volkamer et al., 2005; Chan et al., 2012) and its vertical distribution (Wang et al., 2006), in remote sensing of volcanic emissions (Kern et al., 2009), investigation of atmospheric halogen chemistry in coastal (Peters et al., 2005; Pikelnaya et al., 2007; Keene et al., 2007; Commane et al., 2011), desert (Holla et al., 2015) or polar regions (Hausmann and Platt, 1994;

Hönninger et al., 2004; Frieß et al., 2011; Liao et al., 2011; Stutz et al., 2011), and ship-based in the Arctic sea ice region (Pöhler et al., 2010).

    The main advantage of DOAS in atmospheric remote sensing is that it allows the contact-free and simultaneous measurement of several trace gases. Exploiting that many molecules have unique differential absorption cross sections, mixing ratios can be derived by measuring the optical density of long light paths in the atmosphere using the DOAS principle (see e.g. Platt

and Stutz (2008) for a detailed introduction).

    In contrast to passive instruments (e.g. Multi Axis (MAX)-DOAS or satellite instruments), which use scattered or reflected light and hence rely on natural light sources such as solar (or lunar Wagner et al., 2000) radiation, active DOAS instruments use artificial light sources such as LEDs or arc lamps. The independence from natural light sources allows continuous observations

of trace gases to e.g. study night-time chemistry. It also enables investigations of trace gases absorbing in the deep UV where no natural light sources exist. Another advantage is the well-defined light path of up to 20 km. Along this light path, a mean mixing ratio is determined. In comparison to passive instruments, this reduces the analytical effort to obtain mixing ratios and usually leads to smaller uncertainties as no radiative transport models are needed for the interpretation of the data. Furthermore, compared to point measurements, Long-Path DOAS results are less sensitive to large spatial gradients yielding concentrations

with a better representativeness for comparison with chemsitry models or typical footprints of air borne platforms and satellites.

    Most modern LP-DOAS setups use fibre optics for light transfer between light source, telescope and spectrometer (Merten et al., 2011) and a mono-static telescope, i.e. one telescope is used for both sending and receiving the light reflected from a retro-reflector array. In the following, recent improvements to this setup will be presented which, in combination, can increase

accuracy and hence reduce detection limits of LP-DOAS measurements by a factor of 3 to 4 compared to previous setups. To achieve this, a novel light source type was applied and the light coupling from the light source to the telescope was optimised to reduce stray light. Furthermore, a new configuration of the optical fibres with an improved mode mixing was introduced. In addition to an enhanced measurement performance, these improvements have made the previously quite cumbersome setup



of LP-DOAS instruments considerably easier and now allow the operation of low-maintenance, automated instruments for long-term observations.

In section 2 the state of the art in LP-DOAS instrument design will be described. Improvements of measurement performance and operation procedure following the introduction of the novel light source and changes to the fibre configuration are presented in section 3. In section 4 the influence of residual structures due to fibre modes and a new method for mode mixing to reduce these structures is presented. In section 5 the combined contribution to the improved instrument performance with respect to reduced stray light and reduction of total noise is quantified based both on lab measurements and field campaigns and typical detection limits for setups that incorporate the improvements are presented.

## 2 Long-path DOAS

LP-DOAS instruments couple light from an artificial light source into a telescope which creates a light beam that is transmitted through the atmosphere across a distance ranging from a couple of hundred metres to several kilometres. At the end of this atmospheric path, the light is collected by a telescope and analysed for spectral absorption structures - typically with a grating spectrometer. This originally bi-static setup with separate telescopes for sending and receiving was replaced by a mono-static setup with a single telescope and a retro-reflector array introduced by Axelsson et al. (1990). After reflection at the retro-reflector, the light is received again by the same telescope which reduces the complexity of the setup with regard to power supply and alignment. It also doubles the length of the light path. State of the art LP-DOAS instruments mostly rely on fibre optics for light coupling between light source, telescope and spectrometer (see Fig. 1). Compared to traditional systems that use a complex system of mirrors for the light coupling between light source, telescope and spectrometer, this approach, first introduced by Merten et al. (2011), further reduces the complexity of alignment of the telescope itself and increases the transmittance compared to the coaxial Newton-type telescopes used with the mirror coupling.

### 2.1 State of the art instrument setup

The crucial components in a modern fibre-based LP-DOAS setup are the light source and a Y-shaped optical fibre bundle where one end serves as sending fibre bundle that guides the light from the light source to the telescope and the other end serves as a receiving fibre bundle, leading from the telescope to the spectrometer (see Fig. 1). According to Merten et al. (2011), the fibre bundle in such a "classical" setup (see upper row in Fig. 2 for a detailed schematic of the sections of the bundle) on the transmitting end typically consists of a mono fibre with a large diameter (typically 800 μm) to maximize light collection at the light source (at letter A in Fig. 1 and also column A, upper row in Fig. 2). This fibre is then coupled to a ring of smaller diameter (200 μm) fibres (at letter B/column B) leading to the combined end of the bundle. The monofibre (800 μm) is required to guarantee an equal illumination of all small diameter fibres of the ring. Then the end of the bundle (letter C/column C) is placed close to the focal point of the telescope mirror to create a parallelized light beam. For a fibre at the focal point of a parabolic mirror and omitting beam widening effects, the light emitted by the sending fibre bundle would be imaged on itself,





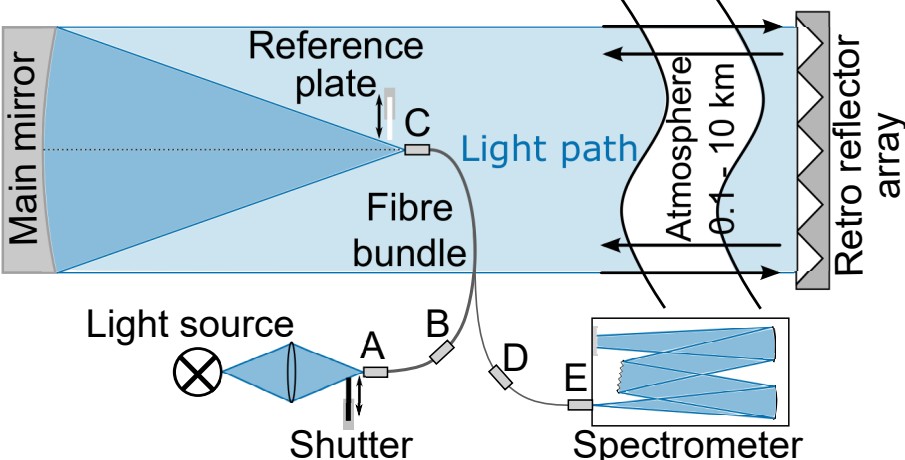

**Figure 1.** Components of a fibre-based LP-DOAS system consisting of a light source, a y-shaped fibre bundle, a telescope for sending and receiving the light, a retro reflector array and a spectrometer.

so that no light would reach the receiving fibre. However, there are a number of effects that blur the reflected image of the light source and lead to a coupling of light into the central receiving fibre: (a) comatic aberration when the incident beam is parallel but not paraxial, (b) diffraction at the apertures of telescope and retro-reflectors (c) surface irregularities of mirror and retro-reflectors, (d) defocussing of the fibre bundle, (e) atmospheric turbulence, and (f) for spherical main mirrors the spherical aberration in combination with the lateral offset of the beam at the retro-reflectors (Rityn, 1967; Eckhardt, 1971; Merten et al., 2011). Merten et al. (2011) have determined (a)-(c) to have a negligible influence for components typically used in LP-DOAS systems. Considering (d) and (e) (and (f) if a spherical main mirror is used), the light throughput of a fibre based system is optimised by setting the end of the fibre bundle to a slightly out of focus position in front of the main mirror.

To homogenize the illumination of the entrance slit of the spectrometer and hence the grating, different mode mixing techniques (see e.g. Stutz and Platt (1997)) can optionally be applied between telescope and spectrometer (at D in Fig. 1/column D upper row in Fig. 2) before the light is coupled into the spectrometer passing an (optional) optical slit (letter E/column E) (see section 4).

For the analysis of the atmospheric absorption, a reference spectrum without absorption of the gases is required. This is obtained by temporarily inserting a diffuse reflector, e.g. a sandblasted white surface, close (i.e. around 1-4 mm distance) to the end of the fibre at (C) thus creating an optical "shortcut" (SC) for the light (for sketch of this mechanism see Fig. A1 in the appendix). In the following, spectra recorded this way will be referred to as reference. The reflector mechanism will be referred to as shortcut.

To account for scattered sun light from the atmosphere in both atmospheric and reference spectra as well as to correct for the CCD's dark current and offset signal, background spectra for both types of measurement spectra are recorded on a regular basis by shutting off the light source at (A). All four spectrum types are recorded in an interleaved fashion with typically a





couple of pairs of reference and atmospheric spectra followed by one atmospheric background and one reference background. Examples of such measurement routines are discussed in detail in section 5.2 below.

## 2.2  DOAS analysis and measurement accuracy

Obtaining average trace gas mixing ratios $\bar{c}_i$ on the light path $L$ between telescope and reflector is based on *Beer-Lambert's law*.

Extended by scattering processes, the attenuation of an initial radiance $I_0(\lambda)$ traversing the atmosphere yielding the measured spectrum $I(\lambda)$ can be described by Eq. 1. The central idea of the DOAS approach is the separation of narrow band (differential) molecular absorption cross section $\sigma_i'(\lambda)$ of a suited absorber $i$ from the broad band portion $\sigma_{B,i}(\lambda)$:

$$I(\lambda) = I_0(\lambda) \cdot \exp\left(-\sum_i [(\sigma_i'(\lambda) + \sigma_{B,i}(\lambda)) \cdot \bar{c}_i \cdot L] - \epsilon(\lambda) \cdot L\right) \tag{1}$$

The optical density $\tau$ (Eq. 2) is calculated taking the logarithm of the ratio of atmospheric $I(\lambda)$ and reference spectra $I_0(\lambda)$

after correcting with their respective backgrounds. Mixing ratios $\bar{c}_i$ are determined by using the differential optical density $\sigma_i'(\lambda)$ modelled from differential literature absorption cross-sections $\sigma_{i,\text{Lit}}'$. To adapt the high-resolution literature cross-section to the resolution of the spectrometer, prior to the analysis it is convoluted with the instrument response function which usually is obtained by recording the shape of an emission line of gas discharge lamp (e.g. mercury). For the detailed mathematical description of the analysis refer to Platt and Stutz (2008). Broad band absorption $\sigma_{B,i}(\lambda)$ in $\tau$ is combined with (broad band)

atmospheric scattering (including both Rayleigh- and Mie-processes) $\epsilon(\lambda)$ and either modelled with a polynomial $P(\lambda)$ or removed with a high-pass filter. The fitting process then minimizes the difference $R(\lambda)$ between measured and modelled optical density using a least-squares approach yielding mixing ratios $\bar{c}_i$:

$$\tau = \underbrace{\ln\left(I_0(\lambda)/I(\lambda)\right)}_{\text{Measurement}} = \underbrace{\sum_i \sigma_{i,\text{Lit}}'(\lambda) \cdot \bar{c}_i \cdot L + P(\lambda)}_{\text{Model}} + \underbrace{R(\lambda)}_{\text{Residual}} \tag{2}$$

The measurement accuracy and hence the detection limit of the retrieved mixing ratios is determined from this difference $R(\lambda)$

called the residual of the DOAS fit. The magnitude of the residual can be affected e.g. by spectral lamp structures, absorbers missing in the model, inaccurate literature cross-sections used in the model, low signal-to-noise ratios or inhomogeneous illumination of the spectrometer grating, where the angular response of the detector causes residual structures (Stutz and Platt, 1997). A fundamental limit is the photon shot noise. Decreasing with the square root of the count number in spectra, its contribution to the residual however, can become very small and residuals attained in this study still were a factor 2-4

larger than pure photon shot noise. To reduce the residual and hence measurement errors and detection limits, in this study the influence of different components of the LP-DOAS instrumental setup was assessed and optimized.

## 2.3  LP-DOAS setups used in this study

We tested several improvements to the classic fibre based DOAS setup (as described in Sec. 2.1), an overview of the three different set-ups is given in Tab. 1 and will be described in the following sections. The systematic comparison of improvements





to the setup was done in the rooftop laboratory of the Institute of Environmental Physics at the University of Heidelberg (setup HD) with an Acton 500i spectrometer and a smaller laboratory telescope that allows quick changes of components but is not suited for outdoor deployment. For atmospheric measurements a 1.55 km light path (one way) passing over a residential area of Heidelberg to another institute was used. Tests with atmospheric measurements were performed during 6 weeks from March

11 until May 3, 2014. In each configuration, measurements were performed for at least 24 h to ensure sufficient statistics and the comparability of different setups.

As for all measurements not performed under fully controlled laboratory conditions, the influence of environmental parameters has to be considered in such a comparison. An important factor in LP-DOAS measurements are variations of the telescope-reflector alignment which can be influenced by changes to the setup as well as environmental parameters such as air

temperatures. To ensure an optimal alignment, as part of the measurement routine and in alternation with measurement periods, an optimisation of the received signal is performed on a regular basis by systematically varying the telescope alignment around the current position and selecting the alignment with the highest signal. LP-DOAS telescopes thus adaptively counter sudden changes to the system transmissivity e.g. through mechanical interaction with the telescope structure as well as long-term drifts.

In addition to the alignment, atmospheric visibility between telescope and reflectors can vary. Potentially very low visibilities

were removed from the comparison data set by excluding days with rainfall. Other visibility conditions with a similar influence (e.g. fog or smog) did not occur during the comparison period.

We estimate the resulting variations of the absolute intensity of the measurement signal from both factors to be 20%. This value has to be considered when comparing absolute atmospheric intensities achieved with the different setups, which determine the temporal resolution of LP-DOAS measurements. For the accuracy, here assessed through the comparison of the

RMS of fit residuals however (Sec. 2.2), due to the use of differential absorption features in DOAS, variations of the recorded absolute intensity only influence photon statistics and hence photon shot noise. Therefore in the comparisons of residuals from atmospheric measurements, the square root of intensity variations has to be considered and an uncertainty of 10% has to be assumed. For the majority of setups tested here, this is much smaller than the systematic differences of residual RMS values between the different configurations.

It should be noted that in contrast to passive DOAS instruments, changes of the global radiation do not affect LP-DOAS measurements since the atmospheric background signal is corrected with regularly recorded background spectra (see Sec. 2.2). Therefore measurements under e.g. overcast conditions can be compared to observations under clear skies.

The combined changes to the LP-DOAS setup, which were found to be the best combination were then tested with a campaign-grade telescope and a smaller Acton 300i spectrometer (setup "NR") during a six weeks campaign in a rural area

in the "Nördlinger Ries" in southern Germany. Findings from both campaigns were incorporated in a new, low-maintenance automated LP-DOAS system (setup "NMIII"). It was operated on the German Antarctic station Neumayer III from January 2016 until May 2018 and allows the assessment of the long-term performance of the different components. All telescopes investigated here, were equipped with spherical, aluminium coated mirrors.



**Table 1.** Overview of the different LP-DOAS setups used in this study.

| Setup | | HD | NR | NMIII |
|---|---|---|---|---|
| Location | | Heidelberg Campus | Nördlinger Ries/Germany | Neumayer III/Antarctica |
| Light source | | various | EQ-99 (2014) | EQ-99X (2015) |
| Fibre bundle | sending bundle length | various | 6 m + 3 m | 8.55 m |
| | receiving bundle length | | 3 m + 7 m | 7.55 m + 1 m |
| | sending fibre diameters | various | 1x200 µm+1x200 µm | 1x200 µm |
| | receiving fibre diameters | | 6x200 µm+1x800 µm | 6x200 µm+1x800 µm |
| | treatment (spectrometer end) | various | 12 µm coarse polished | 5 µm coarse polished |
| Telescope | focal length | 0.6 m | 1.5 m | 1.5 m |
| | mirror diameter | 20 cm | 30 cm | 30 cm |
| | mirror type | spherical | spherical | spherical |
| | Numerical aperture | 0.17 | 0.1 | 0.1 |
| | Telescope front | open | open | covered with quartz glass window |
| Light path | total (one way) | 3.1 (1.55) km | 5.64 (2.82) km | 3.2 (1.55) km/5.9 (2.95 km) |
| Retroreflector | No. of 2″ elements | 7 | 28 | 24 (heated)/(32) |
| | target size (H x W) | 21 x 18 cm | 45 x 32 cm | 60 x 40 cm/(140 x 80 cm) |
| Spectrometer | Model | Acton 500i | Acton 300i | Acton 300i |
| | Focal length | 500 mm | 300 mm | 300 mm |
| | f-number | 6.5 | 4 | 4 |
| | Numerical aperture | 0.07 | 0.12 | 0.12 |
| | Optical slit | 200 µm | 150 µm | 200 µm |
| | Grating | 600 gr./mm 300 nm blaze | 1000 gr./mm holograph. | 1200 gr./mm / 600 gr./mm holograph./300 nm blaze |
| | CCD | Roper Scientific | Roper Scientific | Andor DU440 BU |
| | Spectral resolution | 0.50 nm | 0.49 nm | 0.54 nm/(0.95 nm) |
| | Spectral window | 85 nm | 85 nm | 65 nm/140 nm |





## 3  Light sources and fibre configurations

The light source of a LP-DOAS instrument is a key component because it has a major influence on the achievable signal-to-noise ratio and temporal resolution. The measurement quality depends on both, its temporal and, particularly for arc lamps, the spatio-temporal stability of the light emitting medium (i.e. the plasma) and its spectral characteristics, namely on its (spectral)
radiance (see Platt and Stutz (2008) for comparison of different light sources).

### 3.1  Comparison of light sources

In the past, for most LP-DOAS applications xenon arc lamps have been used that often suffered from poor stability of the light arc which affected the optical coupling into the fibre and hence the effective intensity and shape of the lamp's spectral structures. Furthermore, lifetimes of most models with high radiance were relatively short (200 to 2000 h when in constant use;
Kern et al. (2006)) and regular replacement during longer measurements required - in addition to the considerable expenses - a time-consuming realignment of the optics after each exchange. Depending on the lamp model used, power consumption was high (up to 500 W plus losses in the power supply), which limited the applicability of LP-DOAS instruments. Additionally, the high voltages necessary for ignition are a shock hazard and cause electromagnetic interferences. Although LEDs are useful for compact applications due to their low power consumption and high spatial stability of the light emitting area, up to now light
output is not high enough to achieve sufficient signal-to-noise ratios in the ultraviolet regime below 350 nm. Their application has been so far limited to very compact "single housing" systems and to ensure sufficient spectral stability, often considerable efforts for temperature stabilization are necessary (Kern et al., 2006, 2009; Sihler et al., 2009).

In our new LP-DOAS setups presented here, a novel, commercially available Laser-Driven Light Source (Energetiq EQ-99 and the follow-up model Energetiq EQ-99X, in the following referred to as LDLS) was applied both, for laboratory tests
and different field measurements. Supplying energy to the xenon plasma with an infra-red laser (rather than a high voltage), it combines the advantages of a high power xenon lamp with long lifetime and high spatio-temporal stability of LEDs at a modest power consumption (140 W). Similar to conventional xenon lamps, xenon emission lines, whose differential nature can limit sensitivity in DOAS applications (in particular around 450 nm, a spectral window in which e.g. IO or glyoxal can be detected), are broadened by a high Xe pressure in the bulb. Details on the LDLS can be found in Zhu and Blackborow (2011b); Horne
et al. (2010); Islam et al. (2013)

We couple the light from the LDLS into a fibre with a lens as described in Sec. 3.4 and depicted in Fig. 5. The light source offers the possibility to purge the lamp housing with a constant flow of nitrogen gas. This prevents ozone formation around the light bulb and hence increases output in spectral regions where ozone has absorption bands and reduces the intrusion of pollutants into the lamp housing (see manufacturers' technical notes for details; Zhu and Blackborow, 2011a). In a test we
performed, the radiance at 255 nm increased by about 30 % compared to no purging when a high flow of nitrogen (about 1 L/min) was used. Due to logistical reasons, the LDLS most of the time was purged with filtered and dried air during the measurements reported in this study. In the system used for long-term observations in Antarctica (see Sec. 5.3), the lamp housing was only purged 30 min per day with filtered and dried air attaining a life time of 22500 h.



**Table 2.** Specifications of several lamps used for LP-DOAS measurements.

| Light source | LDLS | XBO-75 | PLI-500 | UV LED |
|---|---|---|---|---|
| Type | Laser-driven LS | Xe-arc | Xe-arc | Light Emitting Diode |
| luminous surface [µm] | 63 x 144 | 250 x 500 | 300 x 300 | 1000 x 1000 |
| (experimental value) | 95 x 146 $\pm$ 6 % | (*) | (*) | |
| Typical lifetime [h] | > 10 000 | 200-2000 | < 200 | > 10 000 |
| Power requirements [W] | 140 | 75 | 500 | 3 |

(*) only valid for new lamps - can increase drastically within days

## 3.2 Adaptation of the optical setup to the light source

In addition to a long life time and high spatio-temporal stability, a further advantage of the LDLS is the very small and stable plasma spot due to the very precise localisation of the plasma inside the bulb in the focal point of the laser. Its dimension in the order of $100\,\mu m$ (full width at half maximum) is about 3 times smaller than in conventional arc lamps (see Tab. 2). This can be exploited in several ways to further improve the design of LP-DOAS systems - first with respect to the configuration of the fibre bundle and overall system optical throughput.

For optical systems in which light propagates unobstructed in a clear and transparent medium, an invariant, the étendue $G$ can be defined as follows (e.g. Welford and Winston, 1978; Markvart, 2007):

$$G = n^2 A\Omega \tag{3}$$

It is the product of the square of the refractive index $n$ of the medium, the area $A$ of the entrance pupil and the solid angle $\Omega$ subtended at this pupil by an object. Since the exact assignment of these quantities depends on the components of an optical setup that are considered, its definition can vary. For the étendue of a light source for example, $A$ could be the size of the emitting area and $\Omega$ the solid angle around the emitter that is covered by the light collecting optics.

The étendue allows to link the spectral radiant flux $\Phi(\lambda)$ (in W) through an optical system with transmittance $\tau(\lambda)$ to the spectral radiance $R(\lambda)$ (spectral radiant flux per solid angle and surface area in $W\,sr^{-1}m^{-2}$) of the light source:

$$\Phi(\lambda) = \tau(\lambda)R(\lambda)G \tag{4}$$

For a system that consists of several components with different étendues, the overall spectral radiant flux $\Phi(\lambda)$ is limited by the component with the smallest $G_{lim}$, which makes it a very useful quantity for optical design considerations. For an optimal overall throughput, the étendues of all components should match as closely as possible.

In fibre-based LP-DOAS setups, typically either the spectrometer (where $G$ is the illuminated area of the entrance slit times the solid angle of acceptance of the spectrometer) or the telescope (where $G$ is the entrance area of the fibre core times the solid angle of the light cone hitting the main mirror) have the limiting étendue (see Tab. 3).





**Table 3.** Étendues for the coupling between the different components in the three setups used in this study (see Tab. 1). For setup HD, the spectrometer étendue for two fibre bundle configurations (classical and reversed) are indicated. See Fig. 2 and Sec. 3.3 for a description.

| Coupling | HD | NR | NMIII |
|---|---|---|---|
| LDLS → fibre | $19.34 \cdot 10^{-4}\,\mathrm{sr\,mm}^2$ | $21.10 \cdot 10^{-4}\,\mathrm{sr\,mm}^2$ | $9.80 \cdot 10^{-4}\,\mathrm{sr\,mm}^2$ |
| Telescope | $28.73 \cdot 10^{-4}\,\mathrm{sr\,mm}^2$ | $9.89 \cdot 10^{-4}\,\mathrm{sr\,mm}^2$ | $9.89 \cdot 10^{-4}\,\mathrm{sr\,mm}^2$ |
| fibre → spectrometer | $5.74 \cdot 10^{-4}\,\mathrm{sr\,mm}^2$ (classical config.)<br>$29.25 \cdot 10^{-4}\,\mathrm{sr\,mm}^2$ (reversed config.) | $76.66 \cdot 10^{-4}\,\mathrm{sr\,mm}^2$ | $76.66 \cdot 10^{-4}\,\mathrm{sr\,mm}^2$ |

For a given light collection solid angle, the LDLS has a small étendue compared to other light sources owing to its small plasma spot. The manufacturer indicates for example a maximum attainable numerical aperture of $NA = 0.447$ (determined by the geometry of the lamp housing) corresponding to a solid angle $\Omega$ of 0.663. Assuming an emitting surface with $100\,\mu\mathrm{m}$ diameter, this yields an étendue $G_{\mathrm{max}} = 52 \cdot 10^{-4}\,\mathrm{sr\,mm}^2$. This is about four times smaller than for a conventional XBO-75

xenon arc lamp with the same coupling optics (see Tab. 2). A small étendue is favourable for optimal utilization of the light source since, regardless of the coupling optics, the usable fraction of the emitted radiation cannot be increased beyond the radiant flux through the element of the system with the limiting étendue.

This is illustrated by an investigation of the coupling between different light sources and fibres with different diameters. For LP-DOAS systems, a light source that efficiently can be coupled into a fibre with a smaller diameter is advantageous because

it allows to use the fibre bundle in a reversed configuration (see Sec. 3.3 and in Fig. 2 lower row).

Using setup HD (see Tab. 1) with a fixed exposure time and number of scans in reference mode, the intensity spectra of several light sources commonly used for LP-DOAS applications were recorded (shown in Fig. 3). Since angular information of the light is lost on the sand-blasted surface of the reference plate, the spectrometer only samples the intensity of the scattered light. For the comparison between light sources and different fibres, therefore only the coupling between the light sources and

the fibre has to be considered.

First, a classical fibre setup described in section 2.1 was used (Fig. 3 panel (a)). The light from the different sources was coupled into a $1\,\mathrm{m}$ fibre with $800\,\mu\mathrm{m}$ diameter that was coupled to a 6x200 $\mu\mathrm{m}$ ring of a $3\,\mathrm{m}$ long y-shaped fibre bundle (see Fig. 2, upper row, 'classical setup'). The receiving fibre was the single $200\,\mu\mathrm{m}$ core fibre of this bundle which was coupled to a $10\,\mathrm{m}$ long $200\,\mu\mathrm{m}$ fibre that led to the spectrometer.

The recorded irradiance of the LDLS in this setup is about twice as high as the conventional xenon arc lamp Osram XBO 75W (in the following referred to as XBO-75), and differential xenon structures are weaker. The commercially no longer available Hanovia PLI-500W (in the following referred to as PLI-500) xenon arc lamp delivers considerably higher irradiances. However, due to poor spatial and temporal arc stability and the resulting introduction of systematic spectral structures as well as a very complicated handling and short lifetime (around $200\,\mathrm{h}$), this light source was excluded from further investigations.

Comparison of the LDLS and high power LED light sources (for the UV a $3\,\mathrm{W}$ Engin LZ1; in the blue spectral range a $3.5\,\mathrm{W}$





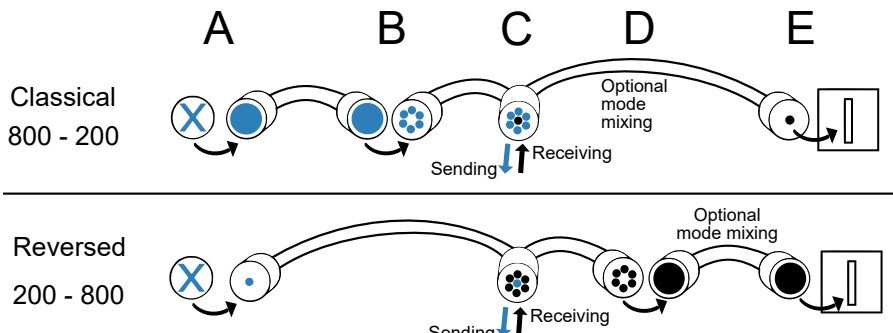

**Figure 2.** Schematic diagrams of sections of the fibre setup and corresponding cross-sections when y-shaped bundle is used in the classical (top row) or reversed (bottom row) way. Additional fibres or fibre bundles can be coupled to the end of the receiving fibre section to improve mode mixing or change the cross section (not sketched). The letters A through E correspond to positions marked in Fig. 1.

Cree XP-E Royal Blue) gives comparable (around $365\,\mathrm{nm}$) or superior irradiances (around $450\,\mathrm{nm}$) for the LEDs however only within the small spectral coverage inherent to the LED principle (Kern et al., 2006).

In a second measurement, a $1\,\mathrm{m}$ long $200\,\mathrm{\mu m}$ diameter single fibre with the same numerical aperture was added to the previous setup between light source and the $800\,\mathrm{\mu m}$ fibre (Fig. 3 panel (b)). This reduces the étendue of the fibre setup by a factor
of 16 (ratio of fibre cross sections). When comparing the effect, it has to be kept in mind that the additional $200\,\mathrm{\mu m}/800\,\mathrm{\mu m}$ fibre interface introduces a coupling loss of about 20-25% (empirical value).

In the setup with added $200\,\mathrm{\mu m}$ fibre (panel (b) in Fig. 3), the smaller étendue of the fibre bundle favours the LDLS with its small, high luminance plasma spot relative to the other light sources. The decrease of the transmitted radiant flux compared to the previous setup (about a factor of 2-3 when correcting for a coupling loss of 25%) and relative to all other light sources
therefore is the smallest for the LDLS. Both, the XBO-75 xenon lamp (reduction by a factor of 9) and the LEDs (reduction by a factor of 11-13) clearly have lower radiant fluxes than the LDLS and even that of the PLI-500 (reduction of a factor of 10) now is only a factor of 2-3 brighter than the LDLS.

### 3.3 Fibre bundle configurations

The favourable properties of the LDLS when coupled into smaller diameter fibres allow to reverse the y-shaped fibre bundle
(see Fig. 2 lower row). The ring of fibres previously used for sending is now used for receiving and the central fibre that previously received the returning radiation now sends it out. Reversing the fibre bundle setup has no influence on the instrument transmissivity since the light path is reversible (see C in Fig. 1 and Fig. 2), but offers several advantages. The larger $800\,\mathrm{\mu m}$ fibre is now coupled to the optical slit. If the limiting étendue of the system is that of the spectrometer (as e.g. in setup HD), a larger illuminated area of the slit increases it (e.g. by a factor of 3.8 when comparing a single $200\,\mathrm{\mu m}$ to the $800\,\mathrm{\mu m}$ fibre with
a $150\,\mathrm{\mu m}$ optical slit). Furthermore, a larger diameter fibre has better mode mixing properties as will be discussed in detail in Sec. 4 below.



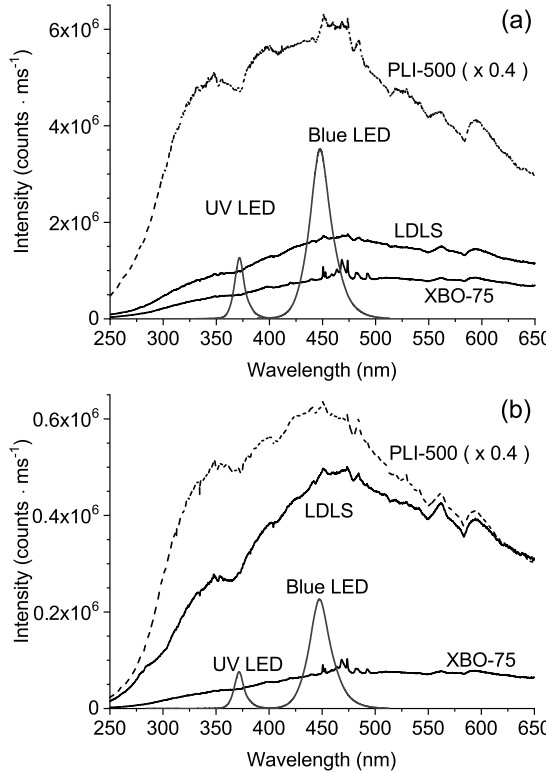

**Figure 3.** Intensities of several light sources measured on the shortcut plate (no atmospheric light path). The position of the lens focussing the light sources onto the fibre was adapted to the respective spectral region. Panel (a): Light coupled into a bundle in classical configuration (see Fig. 2) with 800 µm fibre diameter. Panel (b): Light coupled into the same bundle with an additional 200 µm fibre added in front of the 800 µm fibre of the bundle (also in classical configuration - see Fig. 2). For better comparison, the PLI-500 has been downscaled. Please note that for comparisons between both setups/panels the loss by the additional coupling of the 200 µm fibre, which we estimate to be around 20-25 %, has to be taken into account.

However, the ring of fibres generally has a larger field of view. When in the reversed bundle setup the fibre ring is used for receiving, the atmospheric background light signal (see Sec. 3.4 below) can be six times larger - in particular when the reflector array size is much smaller than the field of view of the ring (often the case in field campaigns due to logistical reasons). This trade-off between the favourable aspects of the reversed fibre bundle configuration like the potential increase in the measurement signal and an improvement of signal quality due to mode mixing (see Sec. 4) has to be weighed against the potentially increased atmospheric background light. If feasible, the latter should be reduced as much as possible by e.g. shielding the area of the ring's field of view around the reflectors with a low reflectance screen.

For applications where radiant fluxes are crucial, a cross-section modulating fibre could be used instead of a large diameter single fibre assuming a detector with sufficient vertical extent. However, it would come at the expense of the favourable mode




mixing properties of larger fibre diameters and can add a potential complication to the determination of the spectrometer's instrument line function which can vary along the optical slit for misaligned fibres in the cross-section modulating fibre or when the different fibres due to atmospheric conditions are not illuminated homogeneously.

In the following sections, the performance of LP-DOAS systems with both, classical and reversed fibre configurations will be compared and discussed (see Fig. 2 upper and lower row respectively). For all classical configurations (Fig. 2 upper row), in the sending section a 800 µm diameter fibre is used to collect the light from the light source, which is then coupled to a ring of 6x200 µm fibres of y-shaped bundles. The receiving section consists of a single 200 µm that is extended by another 200 µm single fibre if necessary. These classical configurations will be denoted "800→200". In all reversed configurations (Fig. 2 lower row), the sending section consists of a single 200 µm fibre at the light source that is either the core of a y-shaped bundle or (if required) an extension fibre then coupled to a core fibre of a y-shaped bundle. The receiving section consists of the ring of 6x200 µm fibres then coupled to a 800 µm diameter single fibre. These reversed configurations will be denoted "200→800".

As the influence of lamp performance and fibre configuration in atmospheric measurements cannot be assessed individually in our setup, the noise of the entire measurement system was investigated to quantify the improvement of measurement quality of this reversed setup and results are discussed in section 5.

## 3.4 Comparison of instrumental stray light

When operating a LP-DOAS, different types of stray light occur that can cause residual structures and limit the measurement accuracy. Atmospheric background light is scattered into the instrument from outside sources (usually the Sun). It depends on the external light source's relative position and orientation as well as atmospheric properties e.g. the visibility. To correct for background light, background spectra for both atmospheric and reference spectra are recorded on a regular basis by blocking the light source (see Sect. 5.2 for details).

Internal or spectrometer stray light is caused by unintended deflections of light inside the spectrometer. In particular in the UV spectral range, where lamp intensities are low compared to the visible spectral range of the light source, this can lead to a systematic underestimation of optical densities since it represents an additive quantity with respect to the total received radiance.

We investigated the amount and origin of spectrometer stray light for different spectral regions using setups HD and NR with the telescope shortcut in place and a set of band-pass filters that block increasing portions of the UV-VIS from 280 nm to 665 nm (see Fig. B1 for the set of filters and their effect on a spectrum). Considering the UV spectral region and the spectrometer of setup HD (f=500 mm) with a 600 grooves/mm grating (blaze 300 nm), stray light levels increase from less than 1% around 400 nm to about 15% at 240 nm. For smaller wavelengths it quickly reaches 80-90% due to the diminishing spectral radiance of the LDLS in this region. About 95 % of the stray light in this configuration originates from the visible spectral range between 400 nm and 650 nm. For a grating with 1200 grooves/mm in the same spectrometer, stray light levels are between 3 to 7 times smaller than in the previous configuration reaching about 2% at 240 nm. For this grating about 50% of the stray light originates from the visible spectral range between 400 nm and 650 nm with the other half being from the




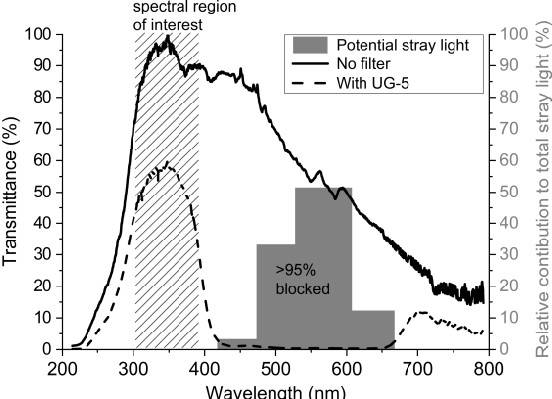

**Figure 4.** Stray light reduction in the UV by using an UG-5 filter. The distribution of the relative contribution of different spectral areas to the total stray light as determined with a set of band-pass filters is shown in grey bars. The two curves show the lamp spectrum determined with setup HD and a classical 800→200 fibre configuration by a measurement on the SC plate with (dashed) and without (solid) a UG-5 band-pass filter between light source and sending fibre.

.

IR. The smaller spectrometer of setup NR (f=300 mm) with a 1000 grooves/mm grating has stray light levels of less than 1% above 320 nm which increases to about 10% at 260 nm.

For evaluations in the spectral region around 330 nm and the 600 grooves/mm (a typical spectral window for the detection of SO2, BrO, formaldehyde, or ozone), an exemplary stray light distribution determined with setup HD with the shortcut in

place is illustrated by the histogram in Fig. 4. The relative importance of stray light for UV measurements is further increased when atmospheric spectra are considered. Since Rayleigh scattering is proportional to $\lambda^{-4}$, loss of UV radiation is higher on the way through the atmosphere compared to the visible parts of the spectrum. This decreases the ratio between UV and VIS and thus increases the relative importance of stray light from VIS spectral regions on UV measurements. Below 300 nm this is further augmented by strong ozone absorption bands. For the NR setup stray light in atmospheric measurements increases

from 1.5% at 320 nm to 10% at 290 nm quickly reaching levels of more than 50% for 280 nm and below.

There are different ways to suppress spectrometer stray light to reduce its influence on the measurement accuracy. One is to add band pass filters (usually colour glass) between light source and fibre to select only the part of the lamp spectrum needed for measurement. In Fig. 4 the effect of a UG5 band pass filter (200-400 nm) on the lamp spectrum is shown. Over 95 % of the light between 400 and 650 nm and hence of the stray light originating from here can be removed while keeping light losses

around 330 nm at about 40-50 % yielding stray light levels of less than 0.1%.

The LDLS with its small and stable arc spot allows a second stray light reduction before coupling the light into the fibre. By mounting the entrance of the fibre (A in Fig. 2) on a stepper motor that can translate around the focal point along the optical axis of the lens which projects the plasma spot onto the fibre end, the chromatic aberration of the lens can be exploited to





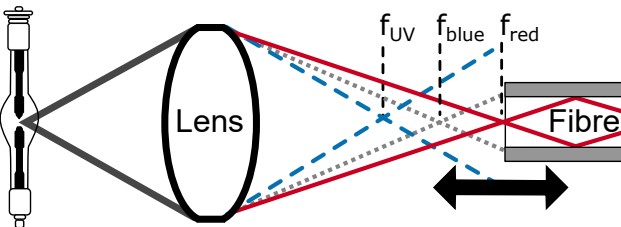

**Figure 5.** Principle of selectively coupling light from a light source into a fibre for stray light reduction using the chromatic aberration of the lens. The foci of different spectral regions can be attained by a translation of the fibre along the optical axis of the lens.

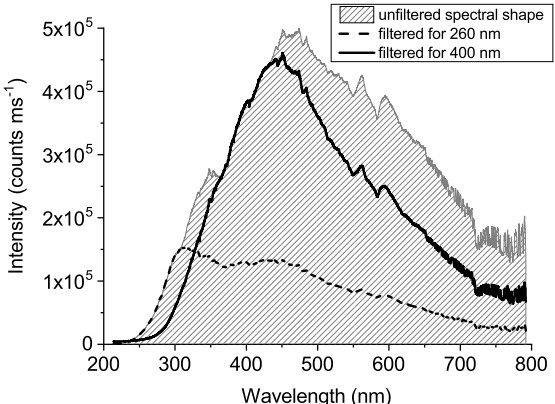

**Figure 6.** Comparison of lamp spectra with the position of the fibre entrance optimized for 260 nm (dashed line) and 400 nm (drawn line) respectively. For comparison a modelled spectral distribution of the light source unaltered by the chromatic aberration of the lens is also shown (shaded area). The latter was obtained by varying the fibre position through all foci from 200 nm to 800 nm and taking the envelope of the resulting spectra.

selectively optimize the input of different spectral ranges (see principle in Fig. 5, see Fig. B2 for a determination of the relative positions of foci for different spectral regions).

Due to the steep increase of the refractive index of quartz glass towards shorter wavelengths (Malitson, 1965), this chromatic filter is particularly selective for the UV spectral range. Figure 6 shows the spectral shapes of the radiation reaching the spectrometer when the position of the fibre is optimised for (a) measurements around 260 nm (dashed line in Fig. 6) and (b) for 400 nm (solid black line in Fig. 6). This can be compared against an artificial intensity distribution that represents the envelope of spectral distributions when the fibre position is tuned through several foci between 200 nm and 800 nm.

When this chromatic aberration filter is optimised for 260 nm, stray light originating around 400 nm, 540 nm, and 680 nm is reduced by about 60 %, 75 %, and 80 % respectively (details are given in appendix B2).




## 4 Fibre modes

The modes of an optical fibre represent different distributions of the light travelling inside the core of the fibre. The solutions of the Helmholtz wave equation for the fibre core using Maxwell's equations and considering core geometry and boundary conditions yield the possible modes (Kaminow et al., 2013). For a given wavelength $\lambda_0$ the number $n$ of modes is proportional to both, numeric aperture $N_A$ and fibre radius $a$:

$$n \propto \frac{a \cdot N_A}{\lambda_0} \tag{5}$$

When the light travels only in a few modes, the resulting light spot leaving the fibre can be inhomogeneous because of the intensity patterns of the individual modes. The intensity distribution between different modes can change along the fibre when energy is transferred from one mode to another which is referred to as mode coupling. This can be caused e.g. by impurities, temperature changes or mechanical stress on the fibre (Stutz and Platt, 1997; Kaminow et al., 2013). In fibres with a smaller diameter in which fewer modes are possible, this inherent or 'natural' mode coupling has a less homogenizing effect than for an otherwise identical fibre with a larger diameter. Therefore, placing a fibre with a larger diameter between telescope and spectrometer as it is the case in the 'reversed fibre configuration' (see Fig. 2 lower row) improves the mode mixing capacity of LP-DOAS setups.

### 4.1 Comparison of mode mixing techniques

In applications with grating spectrometers, an irregular and temporally unstable illumination of the grating resulting from non-uniform illumination of the spectrometer field of view, e.g. due to fibre modes can create systematic, temporally unstable residual structures in the DOAS analysis (see Stutz and Platt (1997) for a detailed study) and thus degrade the measurement accuracy. To reduce these structures, 'artificial' i.e. intentional mode coupling can be induced by different methods. This is referred to as mode mixing (some publications also use the term mode scrambling).

Suitable fibres containing more impurities inherently lead to more mode coupling. However, since impurities cause signal loss when fibres are used in communication applications, manufacturers have improved fibre purities leading to a reduced inherent mode coupling in modern fibres (Kaminow et al., 2013). Since larger diameter fibres allow more modes (Eq. 5), adding such a fibre to the y-shaped bundle in front of the spectrometer can have a homogenising effect. The same is achieved by adding diffusing discs to the optical setup (at the expense of transmissivity). Furthermore, mode coupling by mechanical stress can be induced artificially by squeezing or bending the fibre (micro-bending, e.g. Blake et al., 1986; Stutz and Platt, 1997). However, this requires bare fibres to ensure the transmission of the pressure, which makes the handling quite delicate. The mode mixing is also difficult to reproduce and easily influenced by environmental factors such as temperature. Stutz and Platt (1997) solved this by mechanically vibrating a coiled section of the bare fibre in front of the mode mixer to temporally average over different mechanical conditions. When the intensity of the fibre vibration is increased, micro-bending and temporal averaging are effectively combined and can be applied to fibres with a protective coating. This was done in this investigation by attaching the fibre to a vibrating filter pump.





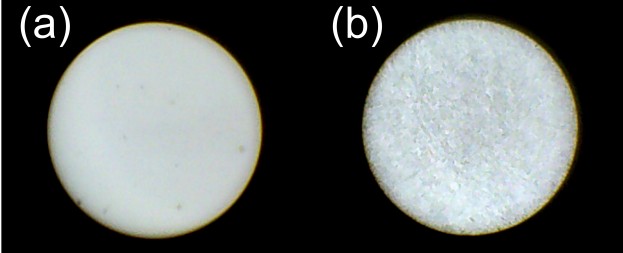

**Figure 7.** 800 µm fibre under the microscope with 30-times amplification. (a) untreated (factory-polished) (b) roughened (12 µm sheet).

A new method we tested for this study is the intentional degrading of fibre ends to create a quasi built-in diffusing disc. To achieve this, fibre ends were treated with polishing sheets - first with 5 µm and then 12 µm granulation. A homogeneous treatment of the surface was insured by visual inspection using a fibre microscope. Figure 7 shows microphotographs of a fibre end surface before and after the treatment.

This new mode mixing approach was compared to the previously used techniques in a series of atmospheric measurements over a 1.55 km light path (one way) using setup HD (see Tab. 1) and the LDLS as light source. All methods were applied to the fibre bundle between light source (A in Fig. 1) and telescope (C) as well as between telescope (C) and spectrometer (E). However, mode mixing between light source and telescope had almost no effect which probably is due to the fact that homogenized light sent out into the atmosphere still can selectively induce modes in the fibre(s) leading to the spectrometer.

One reason for this can be the inhomogeneous illumination of the telescope's field of view because the retro reflector elements do not entirely cover the surface of the array or the light beam partly misses the array. Therefore, in the following only mode mixing between telescope and spectrometer is considered. Light losses caused by the different methods were also quantified and both, the classical and the reversed fibre configurations were tested (Sec. 3.3). In the former setup, a 200 µm fibre was added between the single 200 µm core fibre and the spectrometer. For the latter, a 800 µm fibre was used to couple the 6 x

200 µm fibre ring to the spectrometer. The residual was determined from a fit of 500 added scans between 313 nm and 325 nm considering the cross-sections of $O_3$ (Bogumil et al., 2003), $NO_2$ (Bogumil et al., 2003), HCHO (Meller and Moortgat, 2000), and $SO_2$ (Bogumil et al., 2003). Results for both setups are shown in Tab. 4.

Taking the atmospheric intensity of the uninfluenced fibre as a fixed reference, vibrating the fibre (in the following indicated by "V") causes the smallest light losses for both configurations followed by the novel "Roughened fibre end mode mixing

method" (in the following indicated by "R", 12 µm grit). The diffuser (denoted by "D") leads to light losses of 90% (classical setup) and 75% (reversed setup) respectively.

The roughened fibre end yields the lowest residual RMS values for both fibre configurations. Vibrating the fibre in the classical configuration yields comparable residuals at almost twice the intensity of the roughened fibre end. However, when the fibre is reversed, the roughening yields the overall smallest residuals and only a modest loss of intensity (38% compared to the

uninfluenced fibre against 23% vibrating against the uninfluenced fibre).





**Table 4.** Comparison of different mode mixing methods with LDLS and classical (upper part) reversed fibre configuration (lower part) and their effect on intensity and residual. The number combination in the legend indicates the fibre diameter on the light source end (first number) and spectrometer end (second number). The 800 μm fibre is always coupled to a ring of six 200 μm fibres. Errors of the atmospheric intensities reflect variations through alignment and atmospheric conditions.

| Fibre config. | Method | Atmos. int. (counts ms$^{-1}$) | Residual RMS @ 500 Scans |
|---|---|---|---|
| LDLS-800→200 (classical) | — | $167 \pm 33$ | $14 \cdot 10^{-5}$ |
| | Vibrated (V) | $143 \pm 28$ | $8 \cdot 10^{-5}$ |
| | Diffuser (D) | $17 \pm 3$ | $9 \cdot 10^{-5}$ |
| | Roughened (R) | $83 \pm 17$ | $8 \cdot 10^{-5}$ |
| LDLS-200→800 (reversed) | — | $400 \pm 80$ | $10 \cdot 10^{-5}$ |
| | Vibrated (V) | $310 \pm 62$ | $9 \cdot 10^{-5}$ |
| | Diffuser (D) | $105 \pm 21$ | $7 \cdot 10^{-5}$ |
| | Roughened (R) | $250 \pm 50$ | $6 \cdot 10^{-5}$ |

## 4.2 Temporal stability of mode mixing methods

To reduce measurement errors and to lower detection limits, spectra in the LP-DOAS analysis can be added up before the DOAS fitting process. At the expense of temporal resolution, this reduces photon shot noise. It relies on a high temporal stability of the measurement system. A good mode mixing method therefore should also be temporally stable and residuals should decrease

when spectra are summed (with photon shot noise as fundamental limit). The potential for residual RMS reductions by adding spectra for both, the classical and the reversed fibre configuration was investigated summing spectra over up to 10 hours (see Fig. 8 upper and lower panel). Note that in Fig. 8 results for given measurement times ar plotted, thus the effects of the higher photon shot noise due to signal reduction by the various mode-mixing techniques are included. Averaging measurement data is not necessary for typical applications, but may be required if very weak signals need to be identified.

In the classical fibre configuration, the diffusing disc only attains residual RMS values comparable to vibrated and roughened fibres when 10 hours of observations are added up. Vibrating and roughening for all time spans yields lower results than without additional mode mixing, with lowest residuals for the roughening (Fig. 8, upper panel). In the reversed setup, the differences between the methods are generally smaller which could be due to the favourable effect on mode mixing of the 800 μm fibre coupling into the spectrometer. Residuals for measurements with the diffuser after 60 minutes are even smaller than for a

vibrating fibre. The consistently overall best results for both fibre configurations and all tested mode mixing methods are attained by the reversed (200→800) configuration with roughened fibre end.



### 4.3 The optimal mode mixing setup

Considering light losses (looking at counts per milli second, see Tab. 4), vibrating the fibre leads to smaller losses compared to roughening the fibre end. In the reversed (200→800) fibre configuration, this disadvantage of the roughening is more than compensated by the smaller residuals and also yields the best results when considering the summation of spectra over longer

time periods. Compared with vibrating and especially bending the fibre (Stutz and Platt, 1997), it has the additional advantage of being very reproducible and limiting mechanical stress on the fibre.

For the laboratory comparison with setup HD and a 3 m 800 μm fibre coupled to a 3 m y-shaped bundle with 200 μm fibres, a 12 μm grit gave the best results. For a long-term LP-DOAS instrument for operation in Antarctica with a 8.55 m fibre bundle in reversed configuration that includes a 1 m 800 μm fibre (setup NMIII), 5 and 12 μm roughening gave comparable results with a

lower light loss for the 5 μm grit size. We conclude that the fibre bundle needs to be optimized for the particular measurement setup it is used with considering the trade-off between homogenization of the field of view illumination and light loss for the intended application.

## 5 Overall improvement of LP-DOAS measurement performance

### 5.1 Intercomparison measurements in Heidelberg

In order to quantify the combined effect of the changes to the fibre-based LP-DOAS setup discussed above, atmospheric measurements with different configurations were performed with setup HD over a residential area of Heidelberg for a period of six weeks from March 11 until May 3, 2014. During this time, the different configurations were each tested for at least one day. The influence of atmospheric conditions as well as comparability and representativeness of these observations were discussed in Sec. 2.3.

The measurements were analysed in a UV spectral window between 300 and 350 nm (where trace gases such as $SO_2$, BrO, formaldehyde, and ozone absorb). As a benchmark, the different improvements were compared against a setup with a XBO-75 xenon lamp in a classical 800→200 fibre configuration.

A fibre bundle with a 3 m 800 μm fibre coupled to a 3 m y-shaped bundle with 200 μm fibres was used in the classical (setup XBO_75-800→200) and reversed (setup XBO_75-200→800) fibre configuration before and after the fibre roughening (R), the

most efficient mode mixing treatment identified above (Sec. 4.3) was applied. The measurement performance achieved with the benchmark setup during this comparison period agrees with results from previous measurement campaigns when comparable components were used. Results are summarized in Tab. 5.

Comparing the two light sources with a classical fibre configuration (setups LDLS-800→200 and XBO_75-800→200 in Tab. 5), the higher intensity and better temporal stability of the LDLS discussed in Sec. 3 are apparent. The average residual

RMS for the LDLS is about 30% smaller while the received radiance is about 10% larger. The mode mixing by roughening of the fibre end ("R") reduces the residuals by approximately a factor of 2 for XBO-75 and LDLS (setups LDLS-800→200-R and XBO_75-800→200-R). In the reversed fibre configuration with roughened fibre end mode mixing (setups LDLS-200→800-R



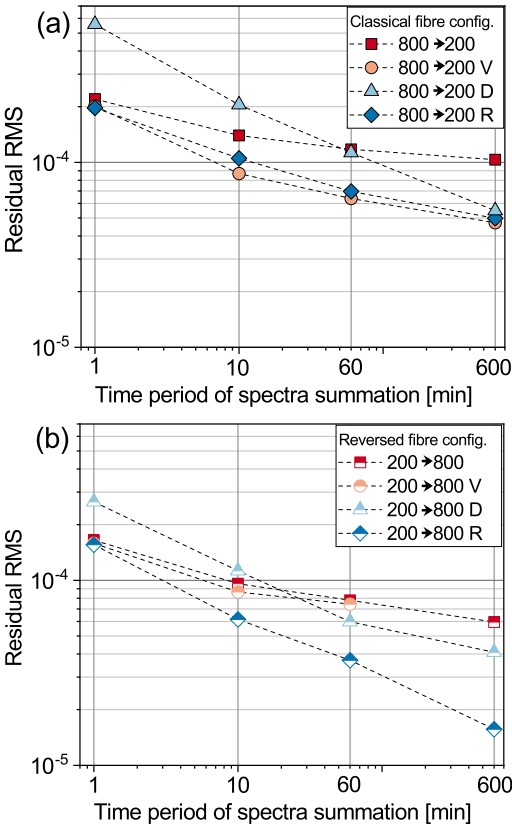

**Figure 8.** Residual comparison for different mode mixing methods and its temporal stability with LDLS and the reversed fibre configuration (panel (a)) and a classical fibre setup (panel (b), for comparison the plot of 'roughened fiber exit' for the reversed setup is also included). The mode mixing methods are abbreviated as follows: V=Vibration; D=Diffuser; R=Roughened fibre end.

and XBO_75-200→800-R), average residual RMS values are comparable. However, the received radiance (and hence temporal resolution for a given signal to noise level) of the XBO-75 is one order of magnitude smaller than for the LDLS, again illustrating the advantage of the smaller arc spot of the latter. Considering the overall improvements from the benchmark setup XBO_75-800→200 to setup LDLS-200→800-R, with reversed fibre configuration and roughened fibre end mode mixing,

5    average residual RMS could be reduced by a factor of 3.5 while detector signals increase by 70%.

## 5.2 Performance in Nördlinger Ries field campaign

Following the investigations in Heidelberg, the modified LP-DOAS setup with the overall best performance (LDLS-200→800-R together with stray light suppression by filters (see Tab. C1 for the models used for the different spectral regions) and additionally using the chromatic aberration approach - see Sec. 3.4 - for further stray light suppression) was deployed with a



**Table 5.** Comparison of atmospheric measurements in Heidelberg utilising the LDLS with different fibre configurations with and without roughening (marked with "R") against the benchmark light source XBO 75 for the HD setup. The number combination in the setup indicates the fibre diameter on the light source end (first number) and spectrometer end (second number). The 800 μm fibre is always coupled to a ring of six 200 μm fibres. The 500 scans cover different averaging times due to different intensities for the configurations but thus contain a similar amount of recorded photons. Errors of the atmospheric intensities reflect variations through alignment and atmospheric conditions.

| Fibre config. | Atmos. int. (counts ms$^{-1}$) | Residual RMS @500 Scans |
|---|---|---|
| LDLS-800→200 | $167 \pm 33$ | $14 \cdot 10^{-5}$ |
| XBO_75-800→200 | $143 \pm 28$ | $21 \cdot 10^{-5}$ |
| LDLS-800→200-R | $83 \pm 16$ | $8 \cdot 10^{-5}$ |
| XBO_75-800→200-R | $40 \pm 8$ | $9 \cdot 10^{-5}$ |
| LDLS-200→800-R | $250 \pm 50$ | $6 \cdot 10^{-5}$ |
| XBO_75-200→800-R | $26 \pm 5$ | $7 \cdot 10^{-5}$ |

campaign grade telescope (setup "NR" in Tab. 1) in a three and a half month measurement campaign from August, 12 until December, 8 2014 in the Nördlinger Ries, a rural area in southern Germany. The campaign had a focus on investigating the link between precipitation and emission of $NO_2$ and HCHO from soils. The LP-DOAS was operated in five spectral windows from the UV to the visible (see Tab. C1) on a 5.7 km light path (2.85 km one way). Since these spectral regions are not

covered at once by the detector, the grating in the spectrometer was turned sequentially to attain the different spectral regions (the measures for stray light reduction i.e. the band pass filters and the chromatic aberration filter approach (see Sec. ) for further stray light suppression in the light source discussed in section 3.4 were changed/adjusted accordingly between spectral windows).

When set up for a particular spectral window, atmospheric and reference spectra were recorded alternately in sets (five

reference spectra interleaved with four atmospheric spectra). At the end of each set, one atmospheric background and one reference background were recorded after blocking the light source. Especially when atmospheric conditions change quickly, short temporal offsets between measurement (atmospheric and reference) and background spectra are important. In each spectral window, several (3-5) of these sets was recorded before the grating was turned to the next spectral position.

To avoid or minimise the influence of detector non-linearity, exposure times for atmospheric and reference spectra in LP-

DOAS measurements were adjusted to yield comparable saturations of the CCD. The exposure times of atmospheric spectra and hence the measurement time were therefore in dependence of atmospheric visibility. The revisiting time of a spectral window (here about 20-30 min) in the whole measurement routine thus depended on the number and total recording durations of the other spectral windows and hence also on atmospheric visibility.

Given a sufficient temporal stability of the instrumental setup, spectra within sets, across sets, or even across subsequent

recordings of a spectral window can be summed to improve the signal to noise ratio and to lower detection limits.



**Table 6.** Attained measurement performance for the detection of different absorbers during the field campaign in the Nördlinger Ries/Germany (NR; total light path: 5.7 km) and during the long term observations on the German Antarctic Station Neumayer III (NMIII; total light paths: 3.1 km and 5.9 km). Were two values are available, attainable residual RMS and detection limits for two different summations and hence temporal resolutions corresponding to 10 min (left value) and 30 min to 2 h (right value) are indicated.

| Absorber | | ClO | $O_3$ | BrO | $SO_2$ | HCHO | $NO_2$ | IO |
|---|---|---|---|---|---|---|---|---|
| Fit range | NR | 290-305 | 295-330 | 290-310 | 320-380 | 320-380 | 420-480 | 410-450 |
| [nm] | NMIII | 287-305.5 | 286.5-329.5 | 302-346 | 286.5-329.5 | 302-346 | 352.5-386.5 | 290-310 |
| Temporal res. | NR | 10* / 120* | 30 | 10* / 120* | 30 | 30 | 30 | 10* / 120* |
| [min] | NMIII | 4 / 40 | 2 / 40 | 2 / 40 | 2 / 40 | 2 / 40 | 2 / 40 | 1.7 / 40 |
| Residual RMS | NR | 20* / 10* | 15 | 15* / 10* | 15 | 15 | 30 | 25* / 15* |
| [$10^{-5}$] | NMIII | 22 / 19 | 28 / 17 | 21 / 13 | 28 / 17 | 21 / 13 | 22 / 13 | 22/ 17 |
| Detection limit | NR | 16.5* / 8.2* | 1124 | 3.7* / 2.5* | 139 | 166 | 61 | 2.1* / 1.2* |
| [ppt] | NMIII | 7.5 / 6 | 420 / 282 | 0.81 / 0.56 | 21 / 14 | 180 / 138 | 62 / 35 | 1.2 / 0.5 |

(*) The halogen-containing molecules were not fitted in the measurements from the Nördlinger Ries, as their presence was not expected. Instead, the respective spectral regions were summed up for 10 min and 2 h and the attained residual RMS values were used to estimate upper limits for the detection limits of these trace gases. See Sec. D in the appendix for details.

For the comparison of instrument performances in different measurement campaigns, no truly regular and universal temporal resolution exists. We therefore indicate typical residual RMS values and detection limits for two temporal resolution regimes: high (2-10 min) and medium (30 min to 2 h). Typical fit ranges, average residual RMS values and detection limits for selected absorbers are given in Tab. 6.

## 5.3 Performance and stability in long-term operation in Antarctica

A second LP-DOAS setup (NMIII in Tab.1) based on the presented improvements was purpose-built for long-term operation on the German Antarctic station Neumayer III (70.67°S 8.27°W) and operated for 31 months from January 2016 until August 2018 (Nasse et al., 2019, , in prep.). It was set up with two light paths 3.1 km (1.55 km one way) and 5.9 km (2.95 km one way) with nearly the same geographical orientation between which the instrument could switch autonomously depending on atmospheric conditions. This was achieved by moving the end of the fibre (C in Figs. 1 and 2) by a motorised x-y translation stage in the focal plane of the telescope main mirror in order to point the light beam to the respective reflector array. Since visibility on the ice shelf at Neumayer III station is often reduced by blowing snow or atmospheric refractions due to strong vertical temperature inversions (optical scintillation and El Mirage effects), most of the time measurements were performed on the shorter light path. The measurement routine was similar to the one applied during the Nördlinger Ries campaign with five different spectral windows (see Tab. C1). Average residual RMS values and detection limits for selected absorbers measured with this setup can be found in Tab. 6.





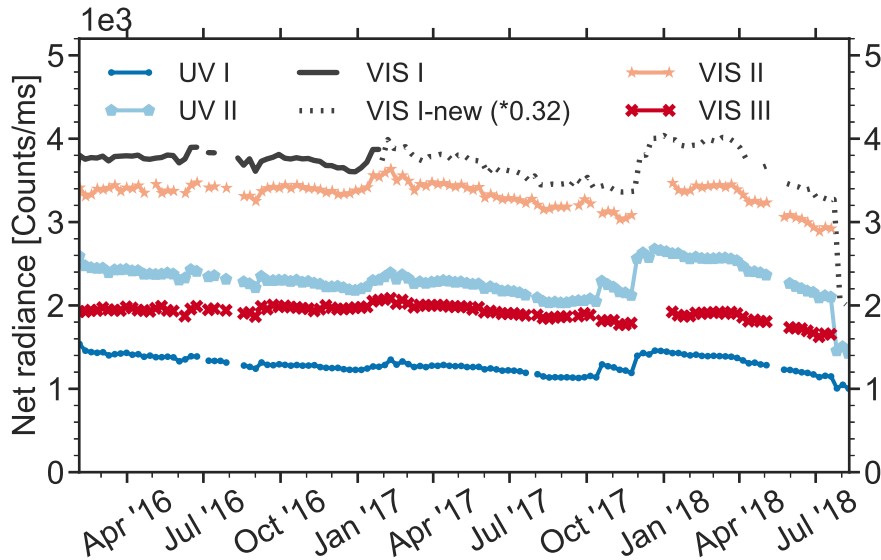

**Figure 9.** Evolution of the net radiance of reference spectra in the NMIII setup over time for the five spectral ranges UV I through VIS II defined in Tab. C1. The average intensity value (in counts) of the reference spectra in the different measurement windows was corrected for background light by subtracting corresponding reference background spectra (light source blocked and shortcut in front of the fibre end in the telescope). Then weekly medians were calculated. Since in the VIS I window, the grating was changed in February 2017, the recorded radiances (which due to the lower dispersion of the new grating were about three times higher) were adjusted to that of the first grating (dotted line). Maintenance was performed on a monthly basis (as far as meteorological conditions allowed) and optical components (filters in the light source) were exchanged after roughly one year of operation. This resulted in temporarily enhanced intensities.

During this long period of continuous operation, maintenance requirements were mostly limited to a monthly cleaning of optical components of the setup, in particular the outside of the quartz front window (added to the Neumayer telescope to prevent snow from entering the telescope and allow the interior heating of the telescope), and regular wavelength calibrations. This routine maintenance and the repair of smaller mechanical malfunctions could be performed by the station's wintering 5 crew. Regular major maintenance was only conducted on a yearly basis.

The long operation time allows to draw conclusions about the long-term performance of the light source and potential ageing effects of the entire setup. Due to logistical reasons and also in view of the low abundances of organics in the air, the LDLS in this setup was only purged daily for 30 min with filtered air (rather than nitrogen gas). Ambient air was passed through a three stage filtering system of silica gel to remove water vapour, active charcoal to remove gaseous pollutants and a particle filter. 10 The light source reached a total operation time of about 22500 hours before a permanent failure occurred. During this time, no realignment of the light source-fibre coupling optics was required which indicates an exceptional spatio-temporal stability of the plasma spot inside the bulb.





It is not possible in this setup to separate contamination and ageing phenomena of the optical components from changes of the radiance of the LDLS. However, an investigation of reference spectra corrected by their respective backgrounds for all spectral windows (Tab. C1) shows similar decreases of the average intensities for all spectral windows (see Fig. 9). The largest decreases of 17% and 12% until the first general maintenance of the instrument after the first year of operation are

observed for the UV I and UV II spectral windows. A thorough cleaning of all optical components which was not always possible in Antarctic winter and an exchange of the band pass filters in the light source could restore intensities to 93% and 95.5% respectively. Throughout the 31 months measurement period, intensities were never lower than 80% of the initial values (except for the days directly before the final lamp failure) even when components probably were dirty. The seemingly irreversible intensity decreases of 4.5-7% for the two UV spectral windows over two years might be explained by a permanent

reduction of the transmissivity of optical components e.g. by solarisation of fibres and lenses or indeed a decreased output of the LDLS.

## 6   Conclusions

Here we present a series of improvements to fibre based long-path-differential optical absorption spectroscopy (LP-DOAS) systems and discuss their respective contributions to the overall improvement of the measurement accuracy. The basis for

this study was a mono-static LP-DOAS setup using optical fibre bundles for light coupling between the different instrumental components.

A laser driven light source (LDLS) with a high-pressure xenon bulb was introduced as a new type of light source for LP-DOAS measurements. It offers a very localised and stable plasma spot with a bright, very weakly structured broad band spectrum that covers a spectral range between 250 nm and the near infrared. When coupled into an 800 μm fibre, as it is

often done in fibre-based LP-DOAS setups, the LDLS intensity is slightly higher than benchmark xenon arc lamps (here a XBO 75W) and comparable to high power LEDs while offering much simpler handling and longer life times than the former (10000 to 22500 h compared to 200 to 2000 h (Kern et al., 2006)) and a better spectral coverage than the latter (see Sec. 3.2). Compared to the XBO 75W arc lamp, which was used as the benchmark in our analysis, employing the LDLS leads to 35% smaller residuals while the received signal and hence the temporal resolution of a setup is about 70% higher.

The small plasma spot of the LDLS allows to reverse the previously used fibre configuration of the LP-DOAS which significantly improves performance (see Sec. 3.3). The larger area of the optical slit now illuminated by a larger fibre can increase the system overall throughput when the spectrometer has the limiting étendue of the system (which in this study is the case for setup HD, see Tabs. 1 and 3). The larger diameter fibre also improves mode mixing and hence reduces irregular illumination of the grating which causes spectral structures limiting the measurement accuracy (see Sec. 4). Alternatively, when light yield

is crucial, the ring of fibres on the receiving side could also be formed into a line resembling the optical slit of the spectrometer further increasing throughput. In atmospheric measurements, these advantages were found to outweigh the potentially higher fraction of atmospheric stray light that can enter the LP-DOAS system in this reversed fibre configuration.





Spectrometer stray light leads to an under-estimation of optical densities and, when spectrally not homogeneous, creates differential structures that limit accuracy. The spectral origin of stay-light was investigated for different spectrometers and gratings. For measurements in the UV and depending on the grating, stray light levels are below 1% for spectral regions above 320 nm increasing to 3 to 15% around 240-260 nm and dominating the signal for lower wavelengths (see Sec. 3.4). Between
50 to 95% of the stray light originates from the visible spectral region between 450 nm and 650 nm. To reduce the influence of stray light, band pass filters adapted to the respective measurement spectral window can be used at the expense of transmissivity (losses of 40-50%), which was already done previously. The application of the LDLS allows a further suppression of stray light by exploiting the chromatic aberration of the quartz lens that couples the light from the light source into the fibre. By precisely positioning the fibre in the focal point of a certain spectral range, light from other, out of focus spectral regions is suppressed.
Combined, these measures lead to an overall stray light reduction of more than 95% for measurements around 330 nm yielding stray light levels of less than 0.1%.

The application of optical fibres in LP-DOAS greatly reduces the complexity of instrument alignment and increases the overall light throughput compared to light coupling with mirrors in the Newton type telescopes used before (Merten et al., 2011). Even though fibres produce a nearly homogeneous illumination, remaining fibre modes can still play a significant role,
e.g. by a slightly inhomogeneous illumination of the spectrometer grating. This causes differential spectral structures limiting the measurement accuracy of DOAS instruments. To homogenise the illumination of the grating, additional mode mixing can be introduced. Here, we compared previously applied techniques like vibrating or micro-bending of the fibre, adding diffusers to the optical setup and a new fibre roughening method (see Sec. 4). In this approach, the highly polished end faces of the fibre bundle are artificially degraded with polishing sheets (5 to 12 μm grit size). Thus a quasi built-in diffuser is added to the fibre.
While the light losses compared to vibrating the fibre are between 20 to 40% higher for the roughened fibre ends, the reduction of the residual is 30% larger when the reversed fibre configuration is considered (Tab. 4). At the same time, the roughening offers the major advantage of being much more reproducible and temporally stable than the vibration. Diffuser disks attain residuals comparable to roughened fibres but light losses are between 4 to 10 times larger which drastically reduces the temporal resolution of measurements. It should be noted that this method is not limited to setups with LDLS but can be
applied e.g. with conventional xenon arc lamps. The residual with a "XBO 75W" and classical fibre bundle with a 800 μm mono- to 200 μm multi-fibre bundle is reduced by a factor of 2, however at the cost of a 70% reduction of the total light throughput (tab. 5).

By combining the changes to the LP-DOAS setup, namely the use of an LDLS with the improved stray light suppression, the reversed configuration of the fibre bundle, and the new mode mixing method, in intercomparison field measurements in
Heidelberg, the residuals could be reduced by a factor of 3-4 compared to the benchmark setup and residual RMS values in the order of $6 \cdot 10^{-5}$ in units of optical density could be achieved (See Fig. 11). Example residual spectra illustrating the improvement are can be found in Fig. 10. As the systematic comparison of the different configurations with benchmark light source and the LDLS for several summation periods in Fig. 11 indicates, this advantage of the new LP-DOAS configuration even increases to a factor of 5 in residual RMS when spectra are summed for 10 h (Fig. 8 and Sec. 5).





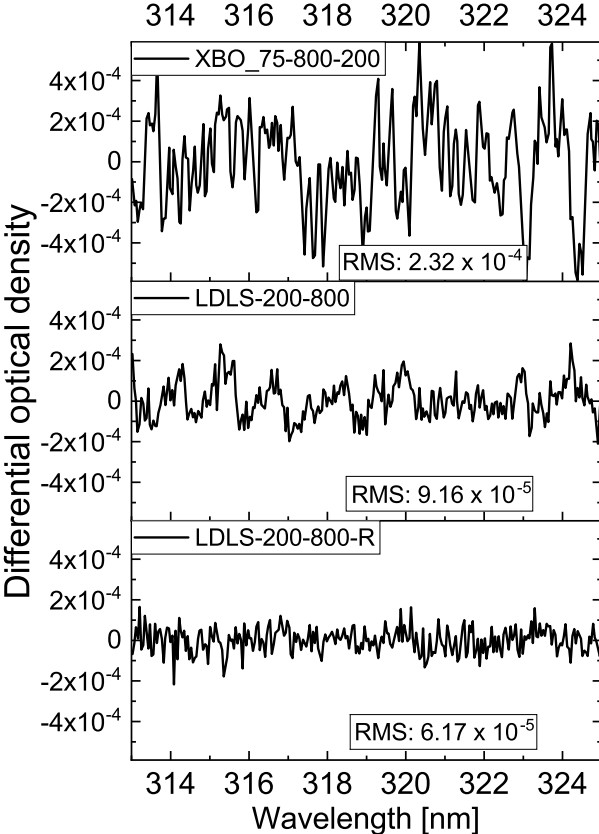

**Figure 10.** Examples of residual spectra and corresponding root mean square (RMS) values for the benchmark configuration with a XBO-75W arc lamp and a classical fibre configuration (top), the new LDLS light source and a reversed fibre bundle (middle) and the new LDLS, reversed fibre bundle and roughened fibre ends (bottom). The latter was found to give best results.

When the improvements described above were applied to two campaign grade LP-DOAS setups using smaller spectrometers (see Tab. 1), residuals of the order of $(0.9 - 1.0) \cdot 10^{-4}$ were achieved under optimal conditions and on average $1.1 - 2.0 \cdot 10^{-4}$ in a long-term measurement campaign in Antarctica. Measurements in the UV spectral region particularly benefit from these improvements. During the measurements in Antarctica, for instance average detection limits of ClO were between 6 to 7.5 pptv at temporal resolution between 4 and 40 minutes. BrO could be detected at detection limits of 0.6 to 0.8 pptv at temporal resolutions of 2 to 30 minutes (see Tab. 6).

In conclusion, the application of the LDLS with its greatly reduced operational complexity and maintenance requirements, its high spatial and temporal stability as well as its long life time has enabled a number of technical improvements to the fibre based LP-DOAS setup. These increase measurement accuracy and reliability of LP-DOAS systems and make this versatile remote sensing technique much easier to deploy even in longer field campaigns or permanently operated applications.





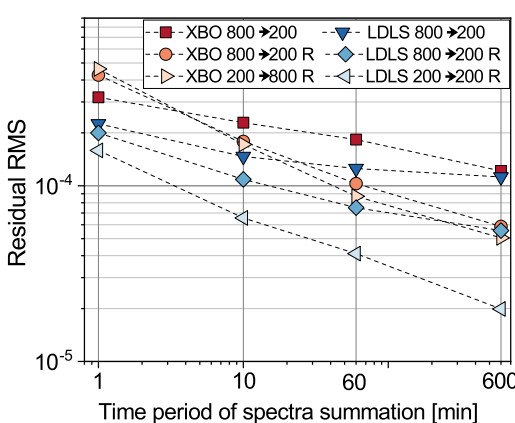

**Figure 11.** Systematic comparison of LP-DOAS setups with classical fibre configuration, classical configuration with improved mode mixing, and the reversed fibre configuration with improved mode mixing for the benchmark light source XBO-75 and the LDLS. Attained residual RMS values for different summation periods up to 10 h are plotted.





## Appendix A:  Additional information on LP-DOAS instrumental setup

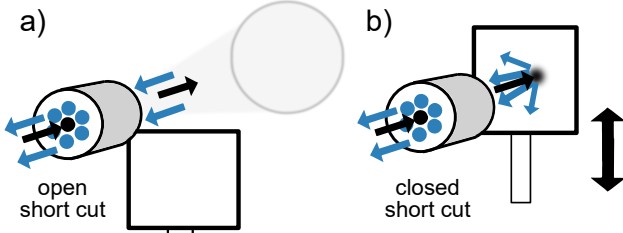

**Figure A1.** Working principle of the shortcut system. When the shortcut is open, light can reach the main mirror, traverse the atmospheric light path and is collected by the fibres (panel a)). To record a reference spectrum, a diffuse reflector plate (here a sandblasted aluminium plate) is moved into the light path at a distance of 1 to 4 mm from the fibre end (panel b)). The radiation is scattered from the surface of this plate back into the fibre bundle without traversing the atmosphere and is thus free of atmospheric absorption.





**Appendix B: Details of stray light investigations**

**B1    Band-pass filters used in stray light investigation**

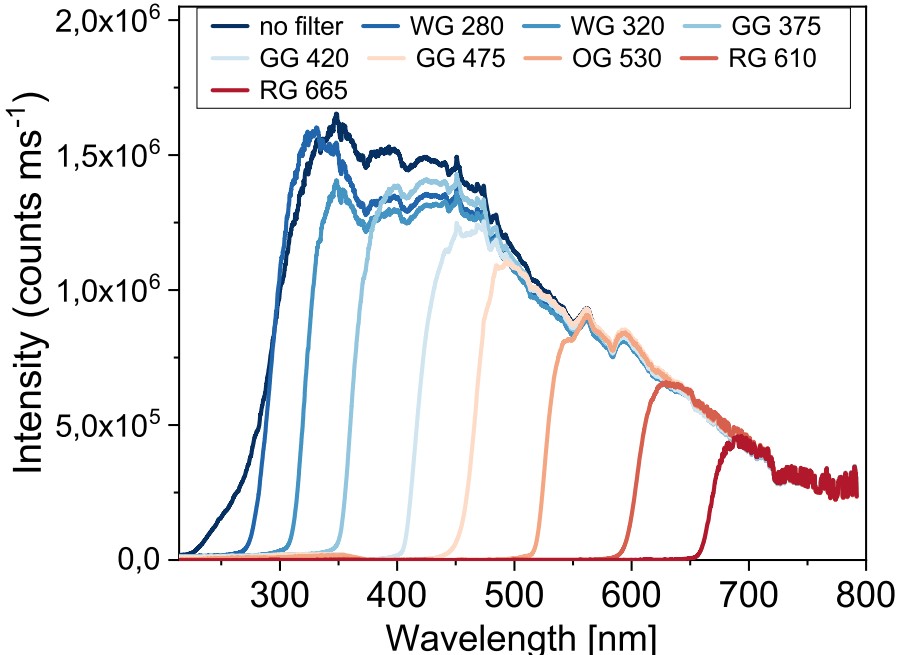

**Figure B1.** Overview of the band-pass and long-pass glas filters used in the investigation of the spectral origin of spectrometer stray light and their influence on the spectrum of a LDLS.



## B2 Locations of foci for different spectral ranges

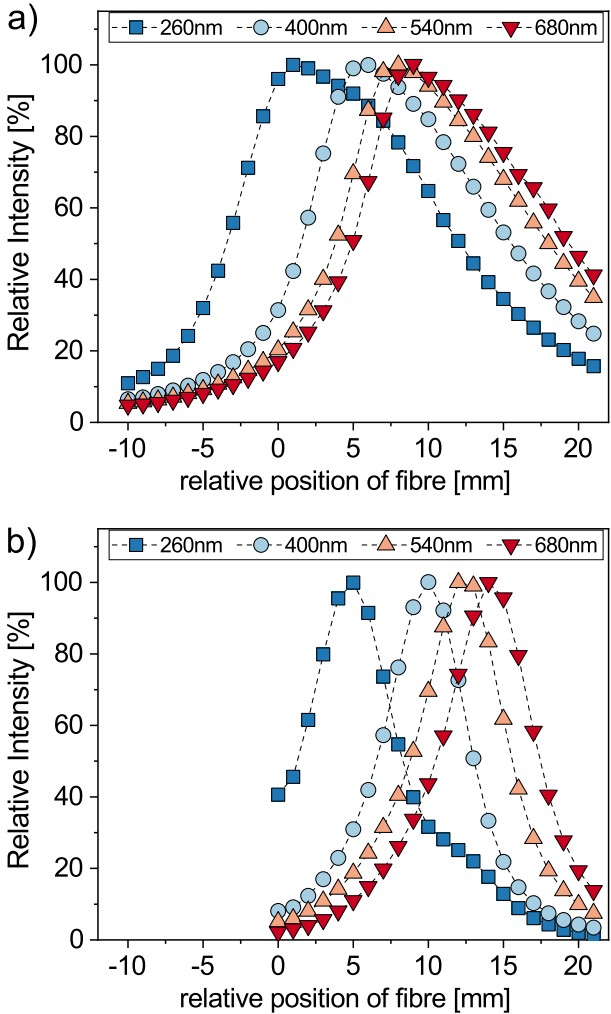

**Figure B2.** Variation of the relative intensity of selected spectral regions as a result of the chromatic aberration as a function of the relative position of the fibre. Panel a) shows results for a coupling into a $800\,\mu m$ fibre, panel b) for a $200\,\mu m$ fibre. For both plot, the increasing selectivity towards the UV due to the non-linear increase of the refractive index is visible. For the smaller fibre diameter the chromatic aberration filter becomes even more selective.



## Appendix C: Measurement routine in field campaigns in the Nördlinger Ries/Germany and Antarctica

**Table C1.** Measurement routines for field campaigns in the Nördlinger Ries/Germany (NR) and long-term observations on the German Antarctic station Neumayer III (NMIII).

|  | Spectral window | UV I | UV II | VIS I | VIS II | VIS III |
|---|---|---|---|---|---|---|
| NR | Spectral range [nm] | 258-343 | 308-393 | 404-489 | 508-593 | 603-688 |
|  | Number of sets | 2 | 5 | 5 | 2 | 2 |
|  | Band pass filter | UG-5 | UG-5 | - | - | OG-550 |
| NMIII | Spectral range [nm] | 280-348 | 327-395 | 378-521* | 528-596 | 614-682 |
|  | Number of sets | 5 | 5 | 5 | 3** | 3** |
|  | Band pass filter | UG-5 | UG-5 | BG-25 | GG-495 | RG-610 |

(*) Due to instrumental stray light problems, another grating was used for the VIS I window in the Neumayer setup (see setup "NMIII" in Tab. 1). (**) For optimisation of the measurement time, these spectral windows were skipped during most of the day time (SZA < 85°) since trace gas molecules of interest here are expected to only be present during the night because of their photochemical instability.





**Appendix D: Estimate of detection limits**

To estimate the expected detection limit of an absorber that is not present in the atmosphere and hence not included in the DOAS fit, the residual RMS value in the spectral region where the absorber would be fitted can be used. A upper limit of a detectable concentration $c_{lim}$ can be inferred by calculating the concentration of the absorber along the light path $L$ for which
5   the optical depth is as large as the root mean square (RMS) of the residual optical density:

$$c_{lim} = \frac{2 \cdot R_{RMS}}{\Delta\sigma' \cdot L} \tag{D1}$$

This estimate tends to be an upper limit for actually achievable detection limits.





*Author contributions.* JMN composed the manuscript and updated all figures based on drafts by PGE (except Figs. A1 and 9). Together with DP, SS, UF, and UP he was involved in the planning, design, and setup of the LP-DOAS instrument in Antarctica and was responsible for its operation. Supported by DP, SS, UF, and UP he analysed and interpreted the Neumayer III data set contributing results in Fig. 9 and Tabs. C1 and 6. PGE, DP, and SS devised the fibre-based optical setup including stray light filtering and fibre treatment used in the lab studies.

5  PE designed, performed, and evaluated all Heidelberg-based measurements supported by DP, SS, and UP. PGE, DP, and SS performed the measurment campaign in the Nördlinger Ries. UF is coordinating the project activites at the German research station Neumayer III.

*Competing interests.* The authors declare no competing interests.

*Acknowledgements.* The measurements at the Neumayer III station were supported by the German Research Association (DFG) in the framework of the project HALOPOLE III (grant number FR 2497/3-2) and by the Alfred Wegener Institute - Helmholtz Centre for Polar

10  and Marine Research. We thank Rolf Weller for advice and practical help during the entire campaign as well as the 36th, 37th, and 38th wintering crews of the station with respective air chemists Thomas Schaefer, Zsófia Jurányi, and Helene Hoffmann for taking good care of the instrument. JMN is supported by scholarships of the Evangelisches Studienwerk Villigst and the Studienstiftung des deutschen Volkes.



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
