# Peer review of "Recent improvements of Long-Path DOAS measurements: impact on accuracy and stability of short-term and automated long-term observations"

_Atmospheric Measurement Techniques, 2019_

## Referee Comment (RC1) · Anonymous Referee #2 · 14 Apr 2019

In their manuscript "Recent improvements of Long-Path DOAS measurements: impact on accuracy and stability of short-term and automated long-term observations", the authors report on some improvements in long path DOAS instrumentation that lead to better measurements and longer instrument life time. More specifically, they describe the use of a new commercial laser driven light source which has a longer life time and creates a more stable and smaller plasma. This can be used to improve instrument throughput by exchanging in- and out-coupling fibres albeit at the price of higher atmospheric straylight. The smaller light spot also allows for a simple method to reduce

instrumental straylight by using the chromatic aberration of the coupling lens to limit the spectral range of the light coupled into the fibre. Finally, the authors test several methods of mode mixing in the fibre including an intentional degradation of the exit surface of the fibres which turns out to create the best results.

The paper is clearly written, reports on some relevant instrumental progress which is of interest to the long path DOAS community and provides demonstration of the value of the improvements on real measurements. I therefore recommend it for publication after minor revisions.

**Detailed Comments**

1. Several places: the term "measurement accuracy" is used in a way which in my opinion is better described by "measurement precision" as absolute accuracy of the mixing ratios is not discussed but rather the reduction in RMS of the residual. Please check and change where appropriate.

2. Page 4, line 2: comatic => chromatic

3. Page 4, line 18: Why is there scattered sun light in short-cut measurements? This should not be the case in my opinion.

4. Equation 1: For sake of consistency, it would be better to denote broad band absorption not with a subscript but with a dash or star like the differential part. Using $\epsilon$ for the scattering cross-section is also maybe not ideal as this would usually be read as extinction which is the sum of scattering and absorption.

5. Page 4, line 13: of gas => of a gas

6. Figure 7: amplification => magnification

7. Figure 8: Why is vibration not resulting in any improvements in reverse geometry, while roughening is very effective?

**[AMTD](top-right)**

**[AMTD](replace)**

8. Section 5.2: I'm not convinced that the reader can learn anything from this section in addition to what is shown in Section 5.3 other than the specific characteristics of a specific LP-DOAS system and would therefore suggest to remove it.

9. Page 21, line 6: section number missing

10. Figure A1: I don't think this figure is needed.

11. Figure B1: If I read this figure right, measurements with filter WG 280 show larger intensities around 300 nm than measurements without filter. This points at the general problems of straylight measurements (stability of lamp and set-up) which should at least be mentioned in the main text.

---

## Referee Comment (RC2) · Anonymous Referee #3 · 22 May 2019

The paper presented by Jan-Marcus Nasse et al. has reported the some recent improvements, i.e. application of a new type of light source and consequent changes to the optical setup, of LP-DOAS measurements to improve the accuracy and stability of short-term and automated long-term observations. In general, it's clear written but not well organized, since the paper length is not in keeping with its importance. I suggest to shorten the paper to some extent. The presented improvements show high attractions to LP-DOAS and related community. It is worthy to be published after some minor revisions.

[Figure]

Specific comments:

P2 L5, I suggest to include some other important atmospheric active species, e.g. HONO, NO3, which made the LP-DOAS technique wide influences in the urban pollution research, along with the related references.

P5 L23-25, Please re-structured this long sentence to be easily understood. "And residuals attained in this study still were a factor 2-4 larger than pure photon shot noise" described the status before improvement or after? Not so clear.

P11 L4, 800 $\mu$m diameter

P18 Line 7, "ar" to "are". In this paragraph, I am not so clear about the "spectra summation". During the experiment, the spectra was sampled in 60s temporal resolution during the 10 hours. The spectra at 10 min, 60 min and 600 min were added with the measured spectra offline? Or the summation was performed during the sampling before the related spectra was recorded.

Fig. 8, Because the LP-DOAS measurements are usually expected to be applied with high temporal resolution e.g. $\sim$ min. Why the authors test with such long periods? I think exam the RMS dependency on the temporal stability of different mode mixing methods should be focused on the even shorter duration of the spectra summation.

P24-27, It's better to shorten the conclusions part, such as Fig. 10, 11 and related detailed description can be moved to the Sect. 5.

---

## Author Comment (AC1) · 13 Jun 2019

We thank Anonymous Referee #2 for the review of the paper and the comments.

The comments of both anonymous referees and our responses as well as a mark up of the changes to the Discussions version of the paper can be found in the attached pdf file in the supplement.

Please also note the supplement to this comment:

[Figure]

https://www.atmos-meas-tech-discuss.net/amt-2019-69/amt-2019-69-AC1-supplement.pdf

---

## Author Response (AR1)

**Recent improvements of Long-Path DOAS measurements: impact on accuracy and stability of short-term and automated long-term observations**

Jan-Marcus Nasse[1], Philipp G. Eger[1, 2], Denis Pöhler[1], Stefan Schmitt[1], Udo Frieß[1], and Ulrich Platt[1]

[1]Institute of Environmental Physics, University of Heidelberg, Im Neuenheimer Feld 229, D-69120 Heidelberg, Germany
[2]now at: Max Planck Institute for Chemistry, Hahn-Meitner-Weg 1, D-55128 Mainz, Germany

*Correspondence to:* J.-M. Nasse (jan.nasse@iup.uni-heidelberg.de)

**Referees' comments and authors' responses**

Page and line indications in the referees' comments refer to the original paper:
https://www.atmos-meas-tech-discuss.net/amt-2019-69/amt-2019-69.pdf

5   The authors' responses refer to this document.

**Anonymous Referee #2**

Comment 1. Several places: the term 'measurement accuracy' is used in a way which in my opinion
10   is better described by 'measurement precision' as absolute accuracy of the mixing ratios is not discussed
but rather the reduction in RMS of the residual. Please check and change where appropriate.
**Response 1:**
The reduction of residual RMS can reflect an improvement of both precision and accuracy. A reduction of
the residual would indeed only affect the precision if the residual would only consist of statistical noise.
15   However, as discussed in the manuscript, there are several effects (lamp instabilities, fibre modes, etc.),
that introduce systematic spectral structures. A reduction of these structures, which have a systematic
influence on the retrieved column densities and hence mixing ratios, improves not only the precision but
also the accuracy. The reduction of systematic structures is the main goal of the improvements presented
here and their result is clearly visible in Fig. 10. Therefore the presented changes lead to an improved
20   accuracy through a reduced influence of systematic structures as well as an improvement in precision.

The difference between accuracy and precision with respect to the residual was discussed in more detail in Sec. 2.2 (P. 12 L. 12 ff) to make this difference clear. The instances when accuracy was used in the original text where checked and where both precision and accuracy are affected/meant, the latter was added (P. 8 L. 26; P. 13 L. 27 and 29; P.32 L. 5, P. 38 L. 4). Where measures against systematic spectral structures are discussed, only accuracy is used.

Comment 2. Page 4, line 2: comatic => chromatic

**Response 2:**

In this list of phenomena that lead to a blurring of the focal point, "comatic" is not a typo but refers to the comatic aberration (coma) that can occur when parabolic telescope mirrors are used and the incident light is not parallel to the optical axis of the mirror.

Comment 3. Page 4, line 18: Why is there scattered sun light in short-cut measurements? This should not be the case in my opinion.

**Response 3:**

To ensure an optimal coupling of light from the sending fibre(s) to the receiving fibre(s), the reference plate is mounted in a distance of 1-4 mm from the fibre bundle depending on the surface properties of the reference plate and the numerical aperture of the fibres (see Fig. A1 for a sketch). This is necessary to ensure a sufficient homogenisation which at shorter distances of the reference plate would be limited by the surface properties of the sand-blasted surface of the reference plate. Since the light path between fibre bundle and reference plate is not shielded from external light, there is a small contribution of background light in short-cut background spectra as well. This contribution is 5-10 times smaller than the background in atmospheric spectra. To correct this but mainly to correct CCD offset and dark current signal of reference spectra, a reference background spectrum is recorded with the light source shut off.

The distance between reference plate and fibres is mentioned in the description of the reference plate system (P. 10 L. 27 of this document). For further clarification, the distance of the reference plate was added to the sketch in Figure A1.

Comment 4. Equation 1: For sake of consistency, it would be better to denote broad band absorption not with a subscript but with a dash or star like the differential part. Using $\epsilon$ for the scattering cross-section

is also maybe not ideal as this would usually be read as extinction which is the sum of scattering and absorption.

**Response 4:**

Equation 1: Similar to the differential absorption cross-section, the subscript "B" was moved into a superscript position. $\epsilon$ in this expression is indeed an extinction (the contribution of Rayleigh and Mie scattering) and not the scattering cross-section. To make clear that the second part of the equation only includes the extinction due to scattering processes, a sum of the two extinction coefficients $\epsilon_R$ for Rayleigh- and $\epsilon_M$ for Mie-scattering was added.

Comment 5. Page 4, line 13: of gas => of a gas

**Response 5:**

The "a" was added to the sentence.

Comment 6. Figure 7: amplification => magnification

**Response 6:**

Amplification was replaced by the correct term magnification.

Comment 7. Figure 8: Why is vibration not resulting in any improvements in reverse geometry, while roughening is very effective?

**Response 7:**

The mode mixing effect of vibrations is a result of the mechanical stress caused on the fibre. Fibres with a larger diameter are less influenced by a given vibration strength than smaller diameter fibres. In contrast, the influence of the roughening is independent of the fibre diameter.

Comment 8. Section 5.2: I'm not convinced that the reader can learn anything from this section in addition to what is shown in Section 5.3 other than the specific characteristics of a specific LP-DOAS system and would therefore suggest to remove it.

**Response 8:**

We agree with the referee's opinion and have moved the discussion of the field campaign in the Nördlinger Ries to the appendix to make the corresponding part of the publication more concise while keeping the

technical details of the LP-DOAS system and typical detection limits in Tables 1 and 6, as this information illustrates that the improvements discussed can be realised independently of specific components such as particular CCD types etc.

Comment 9. Page 21, line 6: section number missing

**Response 9:**

There were two references to the same section in this sentence. One was removed and the other moved within the sentence.

Comment 10. Figure A1: I don't think this figure is needed.

**Response 10:**

In the light of the explanations of the background light in shortcut background spectra (see Response 3), we added the distance between reference plate and fibre bundle to this figure to better illustrate the basic working principle of the reference plate system and therefore prefer to keep the figure.

Comment 11. Figure B1: If I read this figure right, measurements with filter WG 280 show larger intensities around 300 nm than measurements without filter. This points at the general problems of straylight measurements (stability of lamp and set-up) which should at least be mentioned in the main text.

**Response 11:**

The referee's observation in the plot in its current form is correct. The current caption does not sufficiently explain the data shown which leads to an apparently higher intensity of the spectrum with the WG 280 filter. However, it is not the lamp stability that causes the mismatch between the spectrum without filter and the WG 280. To compensate for intensity losses when a filter is placed in the light path (typically around 5-20%), the spectra recorded with a filter in the plot were scaled to match the spectrum without filter in the 700-800 nm spectral region. This leads to the observed apparently higher intensity of the WG 280 spectrum. Additionally, the distance of the fibre from the light source was optimised for a light path with a filter and kept constant (including the measurement without filter). In the measurement without filter the focal point of the lens coupling the signal from the LDLS into the fibre is at a slightly different location which explains the different shape of the spectrum without filter. This is the same effect that is used in the chromatic aberration filter discussed in Sec. 3.4. This information was added to the caption of

the figure.

**Anonymous Referee #3**

Comment 1. P2 L5, I suggest to include some other important atmospheric active species, e.g. HONO, NO3, which made the LP-DOAS technique wide influences in the urban pollution research, along with the related references.

**Response 12:**

The species and according references were added to the list of detectable species.

Comment 2. P5 L23-25, Please re-structured this long sentence to be easily understood. And residuals attained in this study still were a factor 2-4 larger than pure photon shot noise described the status before improvement or after Not so clear.

**Response 13:**

The sentence was restructured into several shorter ones for better comprehensibility. The comparison of attained residuals to the theoretical limit of shot noise underlines the fact that other sources of noise affect the RMS of the residuals. Since the relative importance of shot noise strongly depends on the number of spectra that are summed (i.e. photon statistics), the quantitative comparison of residuals to the shot noise limit was removed. Instead it was made clear that (both before and after the improvements) other sources of noise limit the residual RMS. The reduction of these other factors is achieved through the presented changes to the setup.

Comment 3. P11 L4, 800 $\mu$m diameter

**Response 14:**

The word diameter was added.

Comment 4. P18 Line 7, 'ar' to 'are'. In this paragraph, I am not so clear about the 'spectra summation'. During the experiment, the spectra was sampled in 60s temporal resolution during the 10 hours. The spectra at 10 min, 60 min and 600 min were added with the measured spectra offline? Or the summation was performed during the sampling before the related spectra was recorded.

**Response 15:**

The typo was corrected. The summation in this and all other cases was done after the measurements by combining sets of consecutively recorded spectra into groups corresponding to the different time periods. This was made clearer with an additional sentence detailing the summation procedure.

Comment 5. Fig. 8, Because the LP-DOAS measurements are usually expected to be applied with high temporal resolution e.g. min. Why the authors test with such long periods? I think exam the RMS dependency on the temporal stability of different mode mixing methods should be focused on the even shorter duration of the spectra summation.

**Response 16:**

The goal of the summations over these long time periods was to investigate, whether the different mode mixing techniques would be stable when large numbers of spectra corresponding to long measurement periods are summed up. It is correct that, depending on the absorber, its mixing ratios in the atmosphere, and the optical performance of a setup, LP-DOAS measurements can be performed with a high temporal resolution (e.g. for NO2 under polluted conditions). However, summing up spectra allows to lower detection limits (at the expense of temporal resolution) which is necessary for the detection of molecules with a weaker absorption cross-section or smaller mixing ratio (or both), e.g. detection of reactive halogens in polar environments with mixing ratios down to the sub-ppt range). Prerequisite for this is a sufficient temporal stability of all components (including the mode mixing measures).

Comment 6. P24-27, It's better to shorten the conclusions part, such as Fig. 10, 11 and related detailed description can be moved to the Sect. 5.

**Response 17:**

Following the referee's suggestions, the conclusions were shortened considerably focussing on the quantified improvements presented here. Results illustrating the achieved improvements in comparison with the benchmark setup and in terms of spectral properties of the residual spectra were either moved to section 5.1 (formerly Figs. 10 and 11 now 9 and 10) (Heidelberg intercomparison campaign) and briefly discussed there.

[revised manuscript text omitted]